# Aerobic Exercise Delays Age-Related Sarcopenia in Mice via Alleviating Imbalance in Mitochondrial Quality Control

**DOI:** 10.3390/metabo15070472

**Published:** 2025-07-11

**Authors:** Danlin Zhu, Lian Wang, Haoyang Gao, Ze Wang, Ke Li, Xiaotong Ma, Linlin Zhao, Weihua Xiao

**Affiliations:** 1Shanghai Key Lab of Human Performance, Shanghai University of Sport, 650 Qingyuan Ring Road, Yangpu District, Shanghai 200438, China; zdl981030@163.com (D.Z.); wanglian9369@hotmail.com (L.W.); haoyanggao1999@163.com (H.G.); wz_arrebol@163.com (Z.W.); ley_like@163.com (K.L.); maxiaotong_1026@163.com (X.M.); 2School of Humanities and Social Sciences, Shanghai Lida University, Shanghai 201608, China; 3School of Physical Education, Shanghai Normal University, Shanghai 200234, China

**Keywords:** aerobic exercise, skeletal muscle, Sarcopenia, mitochondrial quality control

## Abstract

**Background:** Sarcopenia is a syndrome associated with aging, characterized by a progressive decline in skeletal muscle mass and function. Its onset compromises the health and longevity of older adults by increasing susceptibility to falls, fractures, and various comorbid conditions, thereby diminishing quality of life and capacity for independent living. Accumulating evidence indicates that moderate-intensity aerobic exercise is an effective strategy for promoting overall health in older adults and exerts a beneficial effect that mitigates age-related sarcopenia. However, the underlying molecular mechanisms through which exercise confers these protective effects remain incompletely understood. **Methods:** In this study, we established a naturally aging mouse model to investigate the effects of a 16-week treadmill-based aerobic exercise regimen on skeletal muscle physiology. **Results:** Results showed that aerobic exercise mitigated age-related declines in muscle mass and function, enhanced markers associated with protein synthesis, reduced oxidative stress, and modulated the expression of genes and proteins implicated in mitochondrial quality control. Notably, a single session of aerobic exercise acutely elevated circulating levels of β-hydroxybutyrate (β-HB) and upregulated the expression of BDH1, HCAR2, and PPARG in the skeletal muscle, suggesting a possible role of β-HB–related signaling in exercise-induced muscle adaptations. However, although these findings support the beneficial effects of aerobic exercise on skeletal muscle aging, further investigation is warranted to elucidate the causal relationships and to characterize the chronic signaling mechanisms involved. **Conclusions:** This study offers preliminary insights into how aerobic exercise may modulate mitochondrial quality control and β-HB–associated signaling pathways during aging.

## 1. Introduction

Sarcopenia, as an age-related syndrome characterized by the progressive loss of skeletal muscle mass and function, not only impairs mobility and increases frailty but is also linked to the development of metabolic disorders [1]. Among the multiple mechanisms implicated in sarcopenia, mitochondrial dysfunction is widely recognized as a key contributor to muscle aging [2]. Mitochondrial quality control, including mitochondrial biogenesis, dynamics, and mitophagy, is critical for maintaining mitochondrial integrity and function. Disruption or imbalance in these processes can lead to mitochondrial dysfunction, thereby exacerbating muscle degeneration during aging [3,4].

β-hydroxybutyrate (β-HB), an important ketone body generated during periods of fasting and aerobic exercise, is increasingly recognized not only as an alternative energy substrate but also as a signaling molecule with important regulatory roles in oxidative stress resistance and mitochondrial function [5,6,7,8,9]. Notably, β-HB has been shown to exert its effects through the activation of the G-protein-coupled receptor 109A(GPR109A), also known as hydroxycarboxylic acid receptor 2 (HCAR2) [10]. Emerging evidence indicates that the β-HB-mediated activation of HCAR2 may modulate the peroxisome proliferator-activated receptor gamma (PPARG) signaling. Both are implicated in metabolic adaptation [11], playing critical roles in modulating energy metabolism and cellular protection [12,13].

In older adults, prolonged sedentary behavior is associated with metabolic dysregulation, impaired muscle protein synthesis, enhanced apoptotic signaling, elevated oxidative stress, and mitochondrial dysfunction. Collectively, these factors contribute to the progressive loss of muscle mass and function, thereby promoting the development and progression of sarcopenia [14,15]. Aerobic exercise is widely regarded as an effective intervention for delaying skeletal muscle aging and promoting mitochondrial health [15,16]. However, the extent to which β-HB, an exercise-induced endogenous signaling molecule, contributes to the protective effects of aerobic exercise against sarcopenia remains unclear. Recent studies have highlighted the emerging role of β-HB and its key metabolic enzyme, D-β-hydroxybutyrate dehydrogenase (BDH1), in exercise physiology. Specifically, aerobic exercise has been shown to upregulate hepatic BDH1 expression, resulting in elevated circulating levels of β-HB [17,18]. Despite these findings, whether exercise-induced endogenous β-HB exerts regulatory effects on skeletal muscle remains unclear, particularly in the context of age-related sarcopenia. The potential role of the β-HB signaling axis in mediating the mitochondrial benefits of exercise in aged muscle has not been fully elucidated. This study aims to explore whether aerobic exercise delays skeletal muscle aging in mice by enhancing mitochondrial quality control, and we hypothesize that this effect may be mediated by the activation of the β-HB/HCAR2–PPARG signaling pathway.

## 2. Materials and Methods

### 2.1. Animals, Study Design, and Ethics

To establish a naturally aged model, we raised 20 male C57BL/6J mice aged 6–15 months. An additional 20 male C57BL/6J mice, aged four months, were obtained from GemPharmaTech Co., Ltd. (Jiangsu, China), to serve as a young control (YC) group. Following the establishment of the aging model, the mice were randomly assigned to one of four groups (*n* = 10 per group): YC, older control (OC), young exercise (YE), and older exercise (OE). All the mice were housed under specific pathogen-free conditions with controlled temperature (22 ± 2 °C, humidity (40–50%), and a 12 h light/12 h dark cycle, with ad libitum access to food and water. All animal experiments were performed in accordance with the guidelines, established by Science Research Ethics Committee at the Shanghai University of Sports (No.102772023DW022) and approved by the Animal Care and Use Committee at the Shanghai University of Sports.

### 2.2. Exercise Protocol

Mice in the exercise groups were subjected to a moderate-intensity treadmill training protocol. During the first week, animals underwent an adaptation phase consisting of five consecutive days of progressive training. Running speed was gradually increased from 10 m/min to 14 m/min, with 20–60 min duration per session. In the second week, the mice engaged in consistent treadmill exercise at a speed of 15 m/min for 60 min per day, 5 days per week, over a period of 16 weeks.

Upon the completion of the exercise intervention, the mice were anesthetized with 1.5–2% isoflurane in oxygen. Under anesthesia, body weight was recorded, and blood samples were collected through cardiac puncture. Serum was isolated by centrifugation and stored at −80 °C for subsequent analysis. The gastrocnemius muscle from one hindlimb was fixed in 4% paraformaldehyde for histological examination, while muscle tissues from the contralateral side were snap-frozen in liquid nitrogen and stored at −80 °C for molecular and biochemical analyses.

### 2.3. Mice Functional Tests

To assess the impact of exercise on muscle function, we measured forelimb grip strength with a grip strength test (YLS-13A, Jinan Yiyan Technology Co., Ltd. Jinan, China), while motor coordination and balance were evaluated using the rotarod test. Grip strength was measured using a horizontally positioned grip strength meter. Each mouse was gently held by the base of the tail and allowed to grasp the metal bar with its forepaws. Once a firm grip was established, the mouse was smoothly pulled backward by the middle of the tail in a straight and continuous motion until it released the bar. The peak force exerted before release was recorded as the measure of forelimb grip strength. Each mouse underwent five consecutive grip strength measurements, and 15 min intervals were implemented between trials [19,20] and normalized to the mouse’s body weight to determine the final grip strength value. The average value of five measurements was recorded as the final grip strength score.

For the assessment of motor coordination and balance, the mice were acclimated to the rotarod apparatus for two consecutive days prior to testing. During this acclimation period, the mice were trained for three minutes per session at a constant speed of 25 r/min. In the formal rotarod test, the mice were placed on a baton-swivel contraption and the duration and speed recorded when each mouse fell from the rotarod. Followed by the testing day (three trials, accelerating 5 to 40 r/min over 5 min), the outcome measure was the best time out of three trials [20,21].

### 2.4. Mice Blood Indexes

Blood samples were collected from the tail veins of mice for the measurement of circulating β-HB levels. Measurements were obtained using a handheld blood ketone meter (Abbott Trading, Shanghai, China) at multiple time points: prior to exercise (baseline) and at immediate times (0 min), 15 min, 30 min, 1 h, 3 h, and 24 h following a single session of aerobic exercise performed at a speed of 15 m/min.

### 2.5. Histological Examination of Gastrocnemius Muscle

Gastrocnemius muscle tissues from each group were fixed in 4% paraformaldehyde (P0099, Beyotime, Shanghai, China) for histopathological analysis. Following fixation, the tissues were embedded in paraffin and sectioned into 4 μm thick slices with a microtome. The tissue sections were dewaxed, rehydrated, and stained with hematoxylin and eosin, and morphological changes in the gastrocnemius muscle were assessed. Stained sections were examined under a light microscope (BX51TF OLYMPUS, Tokyo, Japan) at 400× magnification. For quantitative analysis, five representative images per section were selected and analyzed using Image-Pro Plus 6.0 software (Media Cybernetics, Bethesda, MD, USA). The average cross-sectional area (CSA) of gastrocnemius muscle fibers was calculated by dividing the total muscle fiber area (μm^2^) by the number of individual muscle fibers within each image. The CSA serves as a widely accepted quantitative metric to evaluate muscle fiber size and structural integrity of skeletal muscle.

### 2.6. CAT, GSH Activity, and MDA Content in Gastrocnemius Muscle

CAT activity in the gastrocnemius muscle was measured using a commercial CAT assay kit (A007-1-1, Nanjing Jiancheng Bioengineering Institute, Nanjing, China) following the manufacturer’s instructions. Absorbance values were recorded at a wavelength of 405 nm using a microplate reader (Bio-Rad Laboratories Co., Ltd., Shanghai, China). Similarly, GSH activity was assessed using a GSH assay kit (A006-2-1, Nanjing Jiancheng Bioengineering Institute, Nanjing, China). Tissue samples were processed according to the kit protocol, and absorbance was measured at 405 nm. In addition, MDA content, an indicator of lipid peroxidation, was quantified in gastrocnemius muscle tissues using a lipid peroxidation assay kit (A003-1, Nanjing Jiancheng Bioengineering Institute, Nanjing, China), in accordance with the manufacturer’s guidelines. Lipid peroxidation was assessed using the thiobarbituric acid (TBA) reactive substances assay, which measures MDA levels as an indicator of oxidative damage. Each gastrocnemius muscle tissue sample was homogenized in ice-cold 50 mM phosphate buffer (pH 7.4) at a concentration of 10% (*w*/*v*). The homogenates were then centrifuged at 3000× *g* for 10 min, at 4 °C, and the resulting supernatants were collected. The absorbance of the supernatants was measured at 532 nm using a spectrophotometer. MDA concentrations were calculated based on a standard curve and expressed as nanomoles of MDA per gram of wet gastrocnemius muscle tissue.

### 2.7. Total RNA Extraction and Quantitative Reverse Transcription PCR (RT-qPCR)

Total RNA was extracted from mouse gastrocnemius muscle tissue using TRIzol reagent (15596026, Invitrogen) according to the manufacturer’s protocol. Complementary DNA (cDNA) was synthesized from the isolated RNA using a reverse transcription kit (AG11706, Accurate Biotechnology Co., Ltd., Changsha, China). cDNAs were amplified with Power SYBR green PCR master mix (Q711, Vazyme Biotech Co., Ltd., Nanjing, China) and Prism 7500 instrument (Applied Biosystems). Relative gene expression levels were normalized to β-actin and calculated using the 2^−ΔΔCt^ method. All primers were synthesized by Sangon Biotech (Shanghai, China). Their sequences are provided in Table 1.

### 2.8. Western Blotting

Gastrocnemius muscle samples were first washed with ice-cold normal saline, then lysed in RIPA buffer (P0013B, Beyotime, Shanghai, China) supplemented with protease and phosphatase inhibitors (P1049-1, P1049-2, Beyotime, Shanghai, China). Protein concentrations were determined using a BCA protein assay kit (P0010, Beyotime, Shanghai, China) according to the manufacturer’s protocol. Equal amounts of total protein (20–30 µg) were mixed with 4× SDS-PAGE loading buffer (P0015L, Beyotime, Shanghai, China), heated at 95 °C for 5 min to denature the proteins, cooled to room temperature, and centrifuged at 12,000× *g* for 5 min. Proteins were extracted from whole gastrocnemius muscle lysates without mitochondrial fractionation, and mitochondrial-related proteins were analyzed in the total protein samples. Similarly, for the analysis of signaling proteins (e.g., HCAR2 and PPARG), proteins were measured in whole tissue lysates without subcellular fractionation, reflecting total expression levels rather than compartment-specific distribution. The resulting supernatants were subjected to electrophoresis on 10% SDS-PAGE gels and then transferred onto PVDF membranes (IPVH00010, Millipore, Bedford, MA, USA) using a wet transfer method at a constant current of 220 mA for 1.5 h in transfer buffer containing 20% methanol. A 0.45 μm PVDF membrane was used for proteins larger than 20 kDa, and a 0.22 μm membrane was used for proteins smaller than 20 kDa. After transfer, membranes were blocked with 5% non-fat milk in TBST at room temperature for 1.5 h. For the detection of phosphorylated proteins or in cases where high background was observed, 5% bovine serum albumin (BSA) was used instead of milk. Membranes were then incubated with the appropriate primary antibodies overnight at 4 °C, followed by incubation with HRP-conjugated secondary antibodies for 1 h at room temperature. Protein bands were visualized using an enhanced chemiluminescence reagent (180–505, Tanon, Shanghai, China) and imaged with a gel documentation system (Tanon, Shanghai, China). Band intensities were quantified using Image-Pro Plus 6.0 software, and target protein expression levels were normalized to α-tubulin to account for equal protein loading.

The primary antibodies used in the study were as follows: anti-α-tubulin antibody (11224-1-AP, Proteintech, diluted 1:2000), anti-P70S6K antibody (2708S, CST, diluted 1:1000), anti-P-P70S6K antibody (9234S, CST, diluted 1:1000), anti-4EBP antibody (9452S, CST, diluted 1:1000), anti-P-4EBP antibody (2855S, CST, diluted 1:1000), anti-PGC-1α (66369-1-Ig, Proteintech, diluted 1:5000), antibody anti-Mfn2 antibody (9482, CST, diluted 1:1000), anti-Opa1 antibody (80471, CST, diluted 1:1000), anti-Drp1 antibody (8570, CST, diluted 1:1000), anti-Fis1 antibody (10956-1-AP, Proteintech, diluted 1:1000), anti-BDH1 antibody (ab193156, Abcam, diluted 1:1000), anti-HCAR2 antibody (37602, Signalway Antibody, diluted 1:1000), and anti-PPARG antibody (2435, CST, diluted 1:1000). Secondary antibodies were horseradish peroxidase (HRP)-conjugated goat anti-mouse secondary antibody (abs20039, absin, diluted 1:5000), and goat anti-rabbit secondary antibody (abs20040, absin, diluted 1:5000).

### 2.9. Statistical Analysis

All experimental data were analyzed using SPSS version 25.0, and graphical illustrations were generated using GraphPad Prism 8.0 (GraphPad Software). Data were expressed as mean ± SD. For comparisons among three or more independent groups involving a single variable, one-way analysis of variance (ANOVA) followed by Bonferroni post hoc tests was conducted. To evaluate changes in blood β-HB levels over time and between groups, we used two-way repeated-measures ANOVA to assess group–time interactions. The assumption of homogeneity of variance was tested for all datasets. In cases where this assumption was violated, nonparametric Kruskal–Wallis tests were performed as an alternative. A *p*-value < 0.05 was considered statistically significant for all analyses.

## 3. Results

### 3.1. Aerobic Exercise Mitigates Age-Related Changes in Body Weight, Muscle Function, and Morphology

Aging is known to induce considerable declines in muscle strength, functional capacity, and muscle fiber morphology [22,23,24]. In this study, mice in the OC group demonstrated marked age-related impairments, including a substantial increase in body weight and reduced muscle function, as reflected by decreased absolute muscle weight, decreased muscle-to-body weight ratio, decreased relative grip strength, impaired motor coordination, and smaller CSA of gastrocnemius muscle fibers when compared to the YC group. Notably, aerobic exercise attenuated these aging-associated changes. Mice in the OE group exhibited considerably lower body weight, enhanced grip strength, and improved skeletal muscle mass and CSA relative to OC mice, indicating a protective effect of sustained aerobic training on muscle structure and function during aging (Figure 1A–G).

### 3.2. Aerobic Exercise Promotes Protein Expression of Muscle Protein Synthesis-Related Signaling Pathway Molecules in Aging Mice

Skeletal muscle homeostasis is maintained by a dynamic balance between muscle protein synthesis and protein degradation [25]. In the present study, we observed that, compared with the YC group, the OC group exhibited considerably reduced protein expression of the muscle protein synthesis-related factors eukaryotic translation initiation factor 4E binding protein 1 (4EBP-1) and ribosomal protein S6 kinase (P70S6K). Notably, the expression of 4EBP-1 in the YE group was markedly higher than that in the YC group, suggesting a stimulatory effect of aerobic exercise on protein synthesis even in young mice. Furthermore, compared with the OC group, the OE group displayed substantially increased expression of both 4EBP-1 and P70S6K (Figure 2C,E). Although the phosphorylation levels of 4EBP-1 and P70S6K in the OC group showed a decreasing trend compared to the YC group, and a mild increase was observed in the OE group after aerobic exercise intervention, these changes did not reach statistical significance (*p* > 0.05) (Figure 2B,D).

### 3.3. Aerobic Exercise Enhances the Antioxidant Capacity of Skeletal Muscle of Aging Mice

To evaluate whether aerobic exercise alleviates oxidative stress in aged skeletal muscle, we assessed the activities of CAT, GSH, and the content of MDA. Relative to the YC group, GSH activity in the OC group was considerably reduced, whereas MDA content was considerably elevated, indicating increased oxidative stress with aging. Following aerobic exercise, GSH activity in the YE and OE groups showed a progressive increase, and MDA content considerably decreased, particularly in the OE group (Figure 3B,C). These findings indicate that aerobic exercise enhances the antioxidant capacity of skeletal muscle and mitigates age-associated oxidative damage.

### 3.4. Aerobic Exercise Regulates mRNA and Protein Expression Related to Mitochondrial Biogenesis in Skeletal Muscle of Aging Mice

Moreover, aerobic exercise has been reported to improve mitochondrial function by promoting mitochondrial biogenesis [26]. In this study, aging was associated with considerably decreased mRNA expression of mitochondrial biogenesis-related genes, including *sirt1*, *Nrf2*, *pgc-1α*, and *tfam*. Compared with the YC group, aerobic exercise considerably upregulated the expression of all four genes. Furthermore, relative to the OC group, the OE group exhibited considerably increased expression of *Nrf2*, *pgc*-*1α*, and *tfam*, while *sirt1* expression showed an upward trend that did not reach statistical significance (Figure 4A–D). These results suggest that aerobic exercise partially restores mitochondrial biogenesis signaling impaired by aging.

Consistent with the RT-qPCR results, the protein expression levels of PGC-1α and TFAM were considerably lower in the OC group compared with the YC group. However, aerobic exercise intervention considerably increases the expression of both proteins in the OE group (Figure 4E–G). These findings suggest that aerobic exercise enhances mitochondrial biogenesis and contributes to the improvement of mitochondrial function in the skeletal muscle of aging mice.

### 3.5. Aerobic Exercise Regulates mRNA and Protein Expression Related to Mitochondrial Dynamics in Skeletal Muscle of Aging Mice

As we age, abnormal mitochondrial dynamics may have a negative impact on mitochondrial health [27]. To better understand how mitochondrial dynamics vary in aging skeletal muscle before and after aerobic exercise intervention, we examined the expression of genes and proteins related to mitochondrial fusion and fission. The results of RT-qPCR showed that the mRNA levels of mitochondrial fusion-related factor *mfn1* and mitochondrial division-related factor *fis1* decreased considerably with age in mice, and revealed a decreasing trend in the gene expression of *mfn2* and *drp1*, although these changes were not statistically significant. After aerobic exercise intervention, the mRNA levels of mitochondrial fusion-related factors (*mfn1* and *mfn2*) and mitochondrial division-related factors (*drp1* and *fis1*) were considerably higher in the YE group mice than in the YC group. Furthermore, compared to the OC group, mRNA levels of mitochondrial fusion-related factors (*mfn1* and *mfn2*) and mitochondrial division-related factor *fis1* also increased considerably in the OE group (Figure 5A–D).

Consistent with the results of RT-qPCR, the protein expressions of mitochondrial fusion proteins (Mfn2 and Opa1) and mitochondrial fission proteins (Drp1 and Fis1) in the gastrocnemius of mice decreased considerably with age. However, aerobic exercise increased the protein expression of mitochondrial fusion proteins (Mfn2 and Opa1) and mitochondrial fission proteins (Drp1 and Fis1) in the gastrocnemius of aging mice (Figure 5E–I). These results indicate that mitochondrial dynamics are dysregulated with the aging of mice, which in turn leads to an imbalance in mitochondrial quality control. Aerobic exercise can regulate the mitochondrial dynamics of skeletal muscle in aging mice, and thus contributing to improved mitochondrial quality control of skeletal muscle in mice.

### 3.6. Aerobic Exercise Upregulates Blood β-HB Level in Aging Mice

It is well established that β-HB can be utilized as a nutritional supplement to mitigate age-related diseases by elevating circulating β-HB levels [28]. In the present study, a single session of aerobic exercise led to a rise in blood β-HB concentrations in the OE group, peaking at 3 h post-exercise (1.3 mmol/L) (Figure 6). This observation indicates an acute exercise-induced increase in β-HB levels; however, these data reflect short-term responses and do not provide evidence for chronic adaptations resulting from long-term aerobic training.

### 3.7. Aerobic Exercise Promotes the mRNA and Protein Expression of β-HB/HCAR2-PPARG Signaling Pathway in Skeletal Muscle of Aging Mice

The β-HB receptor, known as HCAR2, is a G protein-coupled receptor activated by endogenous ligands such as β-HB [29]. To investigate the potential mechanisms through which aerobic exercise delays age-related sarcopenia, we explored the regulation of the β-HB/HCAR2–PPARG signaling pathway. Our findings demonstrated that aerobic exercise considerably upregulated the mRNA expression of *Bdh1*, *Hcar2*, and *Pparg* in the skeletal muscle of aging mice (Figure 7A–C). In addition, Western blot analysis revealed considerably elevated protein expression of BDH1, HCAR2, and PPARG in the OE group compared to sedentary controls (Figure 7D–G). These results suggest that aerobic exercise activates the β-HB/HCAR2–PPARG pathway, which may play a role in mediating the beneficial effects of exercise on muscle aging.

## 4. Discussion

Taken together, our study found that aging was associated with reduced muscle mass and function, decreased muscle fiber CSA, imbalanced protein homeostasis, increased oxidative stress, and impaired expression of mitochondrial quality control-related factors. Notably, 16 weeks of aerobic exercise attenuated these aging-related changes and was accompanied by enhanced activation of the β-HB/HCAR2–PPARG signaling axis, indicating a potential mechanistic link between this pathway and the protective effects of aerobic training on skeletal muscle during aging.

Sarcopenia is characterized by an age-related decline in skeletal muscle strength, mass, and function [30]. Previous studies have demonstrated that aerobic exercise can attenuate this degenerative process [31,32,33]. For example, Liu et al. [34] reported that eight months of aerobic training at 75% VO_2_ max considerably improved gastrocnemius muscle mass, strength, and muscle fiber CSA in aged mice. Consistent with these findings, our study showed that aerobic exercise enhanced muscle phenotype and strength in aging mice. However, a notable limitation of the present study is that only the muscle-to-body weight ratio was employed as a relative indicator of muscle mass. Fiber-type composition was not assessed due to constraints in the experimental design. Incorporating these additional parameters in future studies will facilitate a more comprehensive evaluation of sarcopenia-related changes and the therapeutic effects of aerobic exercise.

During aging, muscle protein homeostasis becomes increasingly disrupted due to an imbalance between protein synthesis and degradation [35,36]. Aerobic exercise has been shown to enhance protein synthesis, potentially via activation of the AKT/mTOR signaling pathway [37]. This pathway plays a critical role in regulating muscle protein synthesis, and its downstream targets 4EBP-1 and P70S6K participate in the regulation of muscle mass; associated protein expression and phosphorylation increase after exercise [38]. In the present study, we observed that aging was associated with decreased total protein expression of 4EBP1 and P70S6K. However, 16 weeks of aerobic exercise considerably increased the total expression levels of these proteins in aging mice. By contrast, the phosphorylation levels of 4EBP1 and P70S6K showed only modest increases and did not reach statistical significance. These findings are consistent with previous reports in aged humans, which also failed to detect considerable changes in phosphorylated P70S6K after aerobic training [39]. The discrepancies in phosphorylation responses across studies may be attributed to variations in exercise modality, intensity, and duration [40]. Collectively, our results suggest that aerobic exercise enhances muscle protein synthesis-related signaling, thereby contributing to the attenuation of age-related sarcopenia. It should be noted that the phosphorylated and total proteins were detected on separate membranes, which may cause inconsistencies in band intensity due to variations in antibody sensitivity, membrane conditions, or exposure times. As a result, some calculated p/total ratios exceeded 1. This methodological limitation has also been reported in other studies [41,42] and should be addressed in future research by detecting phosphorylated and total proteins on the same membrane or using multiplex fluorescent detection to improve quantification accuracy.

Aging is also characterized by elevated oxidative stress, largely because of increased production of reactive oxygen species (ROS) and impaired redox homeostasis. This contributes to muscle atrophy and functional decline [43,44]. Antioxidant enzymes such as CAT and GSH play essential roles in neutralizing ROS, whereas malondialdehyde (MDA) serves as a biomarker of lipid peroxidation and oxidative damage [45,46]. Previous studies have reported that regular exercise improves antioxidant defense mechanisms, enhancing enzymatic activity and reducing oxidative damage associated with aging [47,48]. In our study, aged mice exhibited markedly elevated MDA levels and reduced GSH activity, indicating increased oxidative stress. Importantly, aerobic exercise effectively restored these alterations, reducing MDA content and enhancing GSH activity in skeletal muscle. While the inclusion of oxidative stress markers such as CAT, GSH, and MDA provided insight into the antioxidant response, a more comprehensive assessment of redox status would benefit from the inclusion of additional markers such as superoxide dismutase and glutathione peroxidase. Future studies should address this limitation to obtain a more complete understanding of oxidative stress regulation in sarcopenia and the mechanistic benefits of exercise interventions.

Aging disrupts mitochondrial homeostasis, which is characterized by impairments in mitochondrial biogenesis, dynamics, and mitophagy [49,50,51,52,53]. Previous studies have reported age-associated reductions in the expression of proteins involved in mitochondrial fusion (Mfn2 and Opa1) and fission (Drp1 and Fis1) [54,55,56,57]. In the present study, we observed that aging was associated with the downregulation of genes and proteins involved in mitochondrial biogenesis as well as fusion and fission processes. These alterations may contribute to mitochondrial dysfunction, which in turn plays a critical role in the pathogenesis of sarcopenia. Importantly, aerobic exercise upregulated the expression of these mitochondrial quality control-related markers, suggesting a potential role in preserving mitochondrial integrity during aging. However, it is important to note that direct assessments of mitochondrial function (respiratory capacity, ATP production, or mitophagic flux) were not performed in this study. As such, we interpret these findings as expression-level associations, rather than definitive evidence of functional improvements in mitochondrial performance. Future investigations incorporating functional assays of mitochondrial activity are necessary to clarify the mechanistic role of exercise in modulating mitochondrial quality control during aging. Additionally, in this study, mitochondrial-related protein expression was measured in whole gastrocnemius muscle lysates rather than isolated mitochondrial fractions. As a result, we used α-tubulin as the loading control for normalization. While this approach is common and has been adopted in previous studies [58,59], it may limit the specificity of the results to mitochondrial compartments. Future research should include mitochondrial fractionation and use mitochondrial-specific loading controls (such as COX IV or VDAC) to more accurately assess mitochondrial content and dynamics.

Traditionally regarded as a metabolic substrate, β-HB has recently been recognized as a signaling molecule involved in energy homeostasis, oxidative stress regulation, and metabolic adaptation. An increasing body of research indicates that β-HB exerts a range of beneficial effects, including enhanced exercise performance, anti-inflammatory and antioxidant activity, lipolysis promotion, mitochondrial function improvement, and lifespan extension [17,60,61,62,63], which has the potential to alleviate diseases of the skeletal muscle system [64]. Exercise is a key physiological stimulus for ketogenesis, and in the present study, we observed that aerobic exercise acutely increased circulating β-HB levels. However, this elevation was measured following a single exercise session, and therefore represents an acute response rather than a chronic adaptation. As such, we cannot directly conclude that the observed changes in β-HB are sustained over long-term exercise intervention. Future studies should include measurements of baseline β-HB concentrations before and after chronic training to clarify whether persistent changes in circulating β-HB contribute to the long-term improvements in skeletal muscle mitochondrial quality control and function. β-HB is an energy substrate, and BDH1 is a key enzyme for β-HB metabolism. We found that BDH1 expression was reduced in aged skeletal muscle but considerably increased following aerobic exercise, suggesting an upregulation of β-HB metabolism in response to training. However, these observations represent correlational findings and are not definitive evidence of causal relationships. Similarly, the expression of HCAR2, a G-protein-coupled receptor that mediates β-HB signaling, displayed a parallel trend: downregulated in aging muscle and upregulated by exercise. PPARG is a ligand-activated transcription factor involved in lipid metabolism, inflammation regulation, and oxidative stress responses [12,13]. In the present study, we found that the expression level of PPARG showed a consistent trend with HCAR2. These findings suggest a possible activation of the β-HB/HCAR2–PPARG signaling axis following aerobic exercise. Although our results align with previous studies indicating that β-HB can improve mitochondrial function via histone modification [65], and that HCAR2 may regulate PPARG activity [11], we did not directly assess signaling pathway activation or downstream functional effects. Thus, our data align with these trends but cannot confirm mechanistic interactions. It remains unclear whether βHB–HCAR2 signaling regulates PPARG expression, or alternatively, whether aerobic exercise increases PPARG transcriptional activity through other ligands, leading to HCAR2 upregulation as a downstream target [66]. The observed parallel increases in βHB, HCAR2, and PPARG in this study suggest association but do not establish causality. These two hypothetical models require further experimental validation in future studies. Additionally, it should be noted that in this study, protein expression levels of signaling molecules such as HCAR2 and PPARG were measured in whole muscle lysates without subcellular fractionation. This means that the results reflect total protein levels across all cellular compartments rather than specific localization in membrane, cytoplasm, or nucleus. While this approach is commonly used and provides an overall view of expression trends, it may overlook compartment-specific regulatory changes. Future studies should incorporate subcellular fractionation and use compartment-specific loading controls (e.g., Lamin B1 for nuclear proteins) to gain deeper insight into the localization and functional regulation of these signaling proteins. In conclusion, this study provides preliminary evidence that aerobic exercise modulates the expression of genes and proteins related to mitochondrial quality control and β-HB signaling in the context of aging. However, the lack of mechanistic validation limits our ability to infer causality. Future studies should incorporate mechanistic approaches, including chronic exercise models, β-HB supplementation, and the use of genetic tools, such as knockout mice or in vitro systems, to validate the role of the β-HB/HCAR2–PPARG pathway in skeletal muscle adaptation to aging.

## 5. Conclusions

Aerobic exercise alleviates age-related declines in muscle mass and function, potentially by enhancing protein synthesis, reducing oxidative stress, increasing antioxidant capacity, and modulating mitochondrial quality control imbalance. In addition, we observed increased expression of β-HB/HCAR2–PPARG signaling pathway components after aerobic exercises. Notably, the expression pattern of this pathway showed a consistent trend with the improved mitochondrial quality control profile in the skeletal muscle. These findings suggest that the β-HB/HCAR2–PPARG signaling axis is associated with the beneficial effects of aerobic exercises on skeletal muscle during aging. However, further studies are needed to validate the functional role and causal involvement of this pathway.

## Figures and Tables

**Figure 1 metabolites-15-00472-f001:**
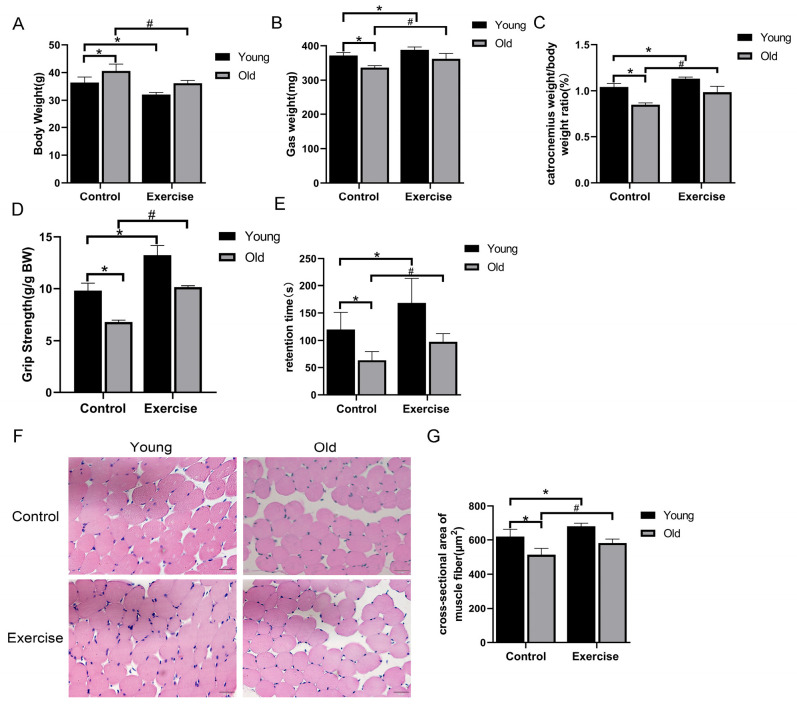
Aerobic exercise improves physical function and muscle morphology in aging mice. (**A**) Body weight. (**B**) Absolute gastrocnemius weight. (**C**) Gastrocnemius weight/body weight ratio. (**D**) Changes in relative grip strength. (**E**) Rotarod test. (**F**) Representative photomicrographs of skeletal muscle sections stained with HE (scale bar, 20 μm). (**G**) Histogram of the CAS of skeletal muscle fibers. All data are presented as mean ± standard deviation (M ± SD) (for panels (**A**–**D**), *n* = 6 mice per group; for panel (**F**), *n* = 3 per group). * *p* < 0.05 vs. YC group; # *p* < 0.05 vs. OC group.

**Figure 2 metabolites-15-00472-f002:**
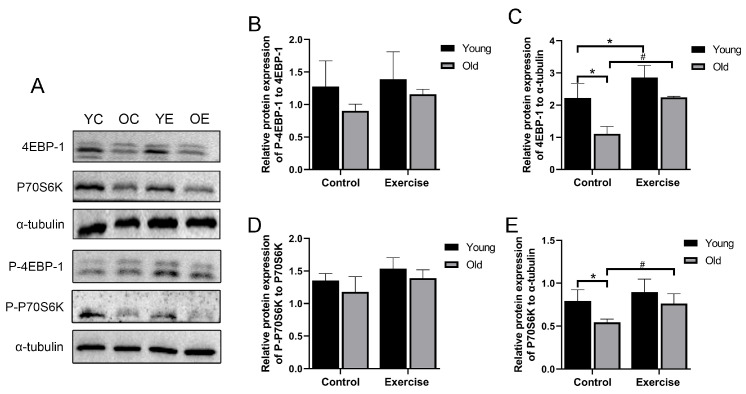
Aerobic exercise improves the muscle protein synthesis in skeletal muscle of aging mice. (**A**–**E**) P-4EBP-1, 4EBP-1, P-P70S6K, and P70S6K protein expression. All data are presented as mean ± standard deviation (M ± SD) (*n* = 3 mice per group). * *p* < 0.05 vs. YC group; # *p* < 0.05 vs. OC group.

**Figure 3 metabolites-15-00472-f003:**
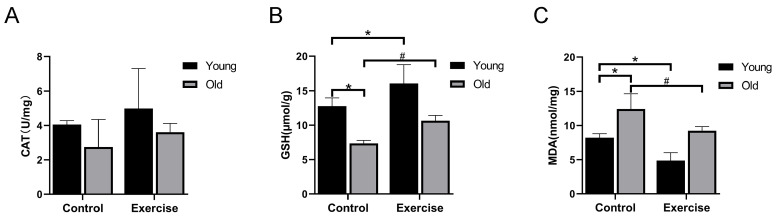
Aerobic exercise protects the skeletal muscle of aging mice from oxidative stress damage. (**A**) The activity of CAT. (**B**) The activity of GSH. (**C**) The content of MDA level. All data are presented as mean ± standard deviation (M ± SD) (*n* = 6 mice per group). * *p* < 0.05 vs. YC group; # *p* < 0.05 vs. OC group.

**Figure 4 metabolites-15-00472-f004:**
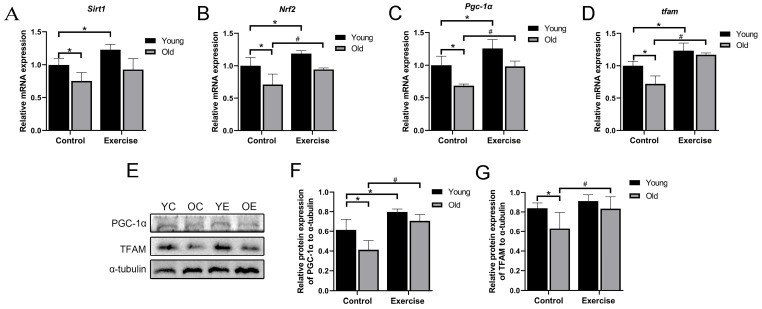
Aerobic exercise improves mitochondrial biogenesis in skeletal muscle of aging mice. (**A**–**D**) mRNA expression of *Sirt*, *Nrf2*, *pgc-1α,* and *tfam* (*n* = 6 mice per group). (**E**–**G**) PGC-1α and TFAM protein expression (*n* = 3 mice per group). All data are presented as mean ± standard deviation (M ± SD). * *p* < 0.05 vs. YC group; # *p* < 0.05 vs. OC group.

**Figure 5 metabolites-15-00472-f005:**
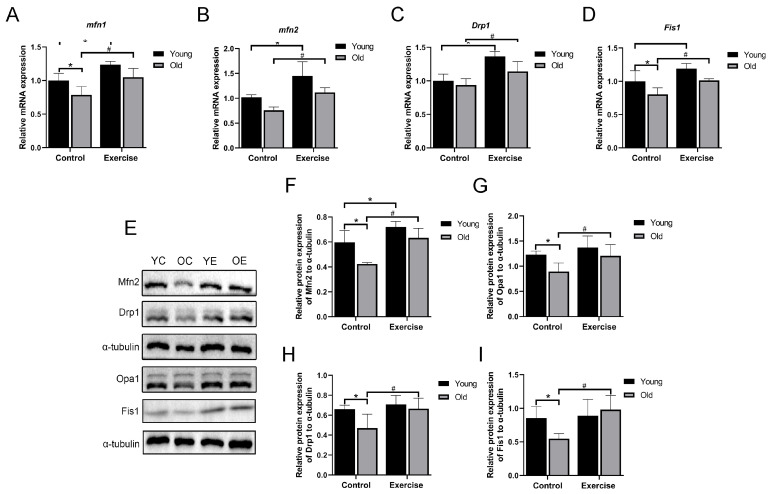
Aerobic exercise improves mitochondrial dynamics in skeletal muscle of aging mice. (**A**–**D**) mRNA expression of *mfn1*, *mfn2*, *drp1,* and *fis1* (*n* = 6 mice per group). (**E**–**I**) Mfn2, Opa1, Drp1, and Fis1 protein expression (*n* = 3 mice per group). All data are presented as mean ± standard deviation (M ± SD). * *p* < 0.05 vs. YC group; # *p* < 0.05 vs. OC group.

**Figure 6 metabolites-15-00472-f006:**
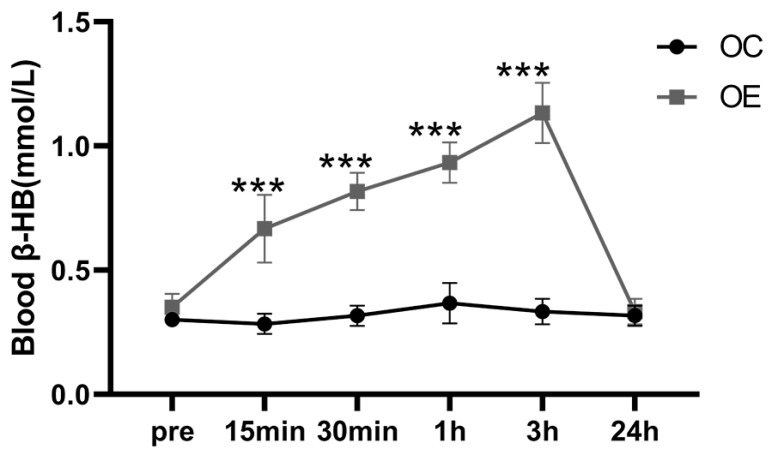
Variation in the blood β-HB levels after a single session of aerobic exercise. All data are presented as mean ± standard deviation (M ± SD) (*n* = 6 mice per group). *** *p* < 0.001 vs. OC group.

**Figure 7 metabolites-15-00472-f007:**
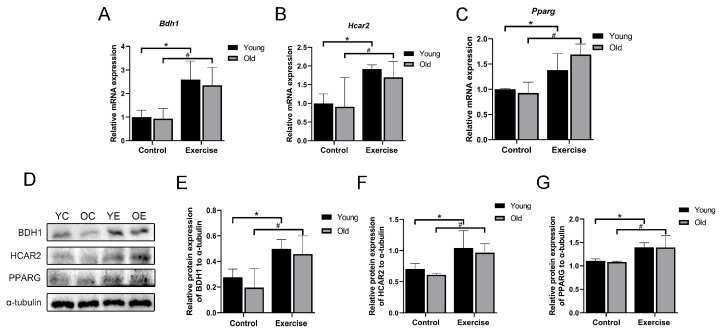
Aerobic exercise activates the β-HB/HCAR2/PPARG signaling pathway. (**A**–**C**) mRNA expression of *Bdh1*, *Hcar2,* and *Pparg* (*n* = 6 mice per group). (**D**–**G**) BDH1, HCAR2, and PPARG protein expression (*n* = 3 mice per group). All data are presented as mean ± standard deviation (M ± SD). * *p* < 0.05 vs. YC group; # *p* < 0.05 vs. OC group.

**Table 1 metabolites-15-00472-t001:** List of primers used for RT-qPCR.

Gene	Forward Primer (5′-3′)	Reverse Primer (5′-3′)
*Nox2*	TGATCCTGCTGCCAGTGTGTC	GTGAGGTTCCTGTCCAGTTGTCTTC
*Nrf2*	TGAGATCTGATGCCTTCTTCTTGCC	CACGACGAGTGTACCTGGGAGTAGC
*Sod1*	TATGGGGACAATACACAAGGCT	CGGGCCACCATGTTTCTTAGA
*Sirt1*	TGATTGGCACCGATCCTCG	CCACAGCGTCATATCATCCAG
*Pgc-1α*	GAAAGGGCCAAACAGAGAGA	GTAAATCACACGGCGCTCTT
*tfam*	AACACCCAGATGCAAAACTTTCA	GACTTGGAGTTAGCTGCTCTTT
*Mfn1*	ATGGCAGAAACGGTATCTCCA	GCCCTCAGTAACAAACTCCAGT
*Mfn2*	AGAACTGGACCCGGTTACCA	CACTTCGCTGATACCCCTGA
*Drp1*	GAAGTGGTGCAGTGGAAATGAC	GTTTCTATTGGGAACCACTGCC
*Fis1*	AGAGCACGCAATTTGAATATGCC	ATAGTCCCGCTGTTCCTCTTT
*Bdh1*	CACCGGAGTGTGTGTAAGGC	CTCGTCTGAACCCGTAGCTC
*Hcar2*	CTGGAGGTTCGGAGGCATC	TCGCCATTTTTGGTCATCATGT
*Pparg*	CCAGGTGACCCTCCTCAAGT	CTGCAGCAGGTTGTCTTGGA
*β-actin*	ACCACACCTTCTACAATGAG	ACGACCAGAGGCATACAG

## Data Availability

The original contributions presented in the study are included in the article; further inquiries can be directed to the corresponding authors.

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
