# Peer review of "Aerobic Exercise Delays Age-Related Sarcopenia in Mice via Alleviating Imbalance in Mitochondrial Quality Control"

_metabolites, 2025, doi:10.3390/metabo15070472_

Round 1

Reviewer 1 Report

Comments and Suggestions for Authors

Review comments:

This manuscript investigates the effects of long-term aerobic exercise on age-related sarcopenia in naturally aging mice, focusing on skeletal muscle mass, function, mitochondrial quality control, and the β-HB/HCAR2-PPARG signaling pathway. While the study addresses an important topic and is grounded in a relevant model, multiple issues must be addressed. Below are specific comments organized by section.

Major:

  1. There are numerous and frequent and awkward or incorrect phrases, inconsistent tense throughout the manuscript. It must be refined for scientific clarity.
  2. Introduction:
  • The introduction jumps across multiple complex topics (inflammation, autophagy, β-HB signaling) without clearly linking them back to sarcopenia or the study aim.
  • Unclear why β-HB is central to this study in introduction. The introduction builds a mechanistic case for β-HB but does not explain why the authors focus on this metabolite over other pathways involved in sarcopenia. And the mechanistic discussion of β-HB, HCAR2, and PPARG is overly detailed for an introduction and interrupts the flow. Consider moving mechanistic detail to the discussion.
  • While mitochondrial dysfunction is presented as a known contributor to sarcopenia, the novelty of linking it to β-HB signaling and aerobic exercise is not clearly positioned in introduction.

Method:

  • What is adaptive training protocol? What speed and duration?
  • The phrase “the mice were anesthetized for weighing and then blood was taken and executed” is unclear and awkward. The authors should clarify the method of blood collection, including how the blood was obtained, processed, and stored after collection.
  • For grip strength, the manuscript does not specify which limb. Forelimb? All-limb?? Grip strength is likely body-weight dependent, however no normalization is mentioned.
  • The phrase “the mice were asked to select the best outcome” is both scientifically inaccurate and grammatically incorrect. The authors should rephrase this and clearly specify which outcome metric was selected for analysis
  • What is “environmentally friendly GD fixative”? The authors should provide the full chemical composition or, if a commercial product was used, the supplier and catalog number.
  • Western blotting: Authors missed details in how they prepared running sample? What is transferring method? Authors states that “transferred onto a 0.45μm or 0.22μm polyvinylidene difluoride (PVDF) membrane”, “Membranes were blocked with 5% bovine serum albumin or non-fat milk at room temperature for 1.5 h”, but it is unclear under which conditions each was used. The authors should clarify the rationale and experimental conditions for selecting membrane pore sizes and blocking agents. Moreover, details regarding the normalization strategy are missing and should be included.
  • Statistical Analysis: “Paired-sample t-tests” is inappropriate unless the same subjects are measured under two conditions, likely not the case here. Authors should re-analyze the data using the correct test. Also, the methods do not specify which statistical test was used for which dataset and there is no mention of post-hoc tests after ANOVA.
  1. Results:
  • A total of nine small figures are presented, many of which are unnecessary. Several figures can be consolidated, for example, Figures 1, 2, and 3 could be combined. The authors should reorganize the figures and related results for conciseness and better readability.
  • The statement, “The results of the mice grip strength test showed that the grip strength of the older control (OC) group mice decreased significantly with age,” is inaccurate. Since the study did not track longitudinal changes in the same animals, but instead compared independent groups, the correct phrasing should be: “The OC group mice showed significantly lower muscle strength compared to the YC group.”
  • Additionally, the sentence “Conversely, it was found that muscle strength was higher than that of the youth control (YC) group mice after aerobic exercise...” is confusing. It is unclear which group showed higher strength than the YC group, YE or OE? The authors should rephrase for clarity.
  • Muscle weight to body weight ratio alone is not ideal, it would be more informative to also include absolute muscle weight and fiber-type analysis.
  • Author states “Compared with the YC group mice, the number of skeletal muscle fibers was significantly decreased in the OC group mice, accompanied by differences in the size, morphology, and number of muscle cells, larger cell spacing, and a decrease in the cross-sectional areas (CSA) of skeletal muscle fibers.” is not supported by data. The authors do not present any quantification of fiber number, size variability, or spacing.
  • In section 3.4, the authors overinterpret trends. For example, non-significant changes in phosphorylation levels are described as biologically meaningful. This weakens the credibility of the conclusions and should be revised.
  • Only 3 markers (CAT, GSH, MDA) are used for measuring antioxidant activity, broader redox panel (e.g., SOD, GPx) would provide stronger support.
  • Single-session β-HB measurement does not directly support the chronic effects discussed throughout.
  1. Discussion:
  • Much of the discussion re-describes findings already stated in the Results section, without deeper analysis or integration with broader literature with minimal critical integration across different domains
  • There are overinterpretation and unsupported claims, for example, “Improved mitochondrial quality control” is frequently used, but no direct assessment of function is presented; Statements like “aerobic exercise reversed the decline of BDH1” imply causality without experimental proof; repeatedly attributes effects to the β-HB/HCAR2/PPARG axis but provides no experimental validation beyond expression changes.

Minor:

  1. R26: “date” is typo, should be “data”
  2. Overuse of "has been shown" or "studies have found" in introduction without critical synthesis of literature.
  3. There are some informal or awkward phrases need to be modified, for example, , “Closely related to the promotion of muscle protein synthesis”, “Exercise-activated endogenous β-HB”…
  4. R141: Acronym “CSA” should be defined when it first mentions.
  5. Catalog numbers should be listed for all commercially purchased supplies or reagents.

Comments on the Quality of English Language

There are numerous and frequent and awkward or incorrect phrases, inconsistent tense throughout the manuscript. It must be refined for scientific clarity.

Reviewer 2 Report

Comments and Suggestions for Authors

This study investigated the important issue on the effects of aerobic exercise on age-related sarcopenia and on mitochondrial quality control in mice. This study provides interesting information. However, the following points should be improved.

  1.    What is new?  As pointed out in Discussion,  many results appear to be consistent with those of previous studies reported. Thus, the originality of this study is unclear.
  2. In Introduction, more detailed information on the effects of exercise on the β-HB/HCAR2-PPARG signaling pathway reported should be provided to make clear what is known and what is unknown in this issue.
  3. To make clear the idea of this study , authors should explain their hypothesis. 
  4. What is the most important impact in this study?  
  5. In this study, some data were obtained with the analysis for only three samples (n=3)/group. Is this sample size enough?
  6. Basic data such as body weight, food intake and blood biochemical parameters relating to energy metabolism, oxidative stress, and inflammation should be indicated.

Round 2

Reviewer 1 Report

Comments and Suggestions for Authors

The authors have addressed all of the major concerns raised in the initial review. For experiments that could not be completed, they have appropriately acknowledged the limitations within the manuscript. Overall, the revisions have significantly improved the quality and clarity of the work. The manuscript is ready for publication, pending correction of the minor issue noted below.

  1. Row 196: authors indicated they used β≤-actin for protein normalization, however, they actually used α-tubulin. β≤-actin should be revised to " α-tubulin"

Reviewer 2 Report

Comments and Suggestions for Authors

This manuscript was carefully and well revised. Hence, this paper is appropriate for publication.
